# Sparse Neural Architectures via Deterministic Ramanujan Graphs

**Suryam Arnav Kalra**                                     *suryamkalra35@gmail.com*
*Department of Computer Science and Engineering*
*Indian Institute of Technology, Kharagpur*

**Arindam Biswas**                                     *arin.math@gmail.com*
*Polynom, Pairs, France*

**Pabitra Mitra**                                     *pabitra@gmail.com*
*Department of Computer Science and Engineering*
*Indian Institute of Technology, Kharagpur*

**Biswajit Basu**                                     *BASUB@tcd.ie*
*School of Engineering*
*Trinity College Dublin, Dublin 2, Ireland*

**Reviewed on OpenReview:** *https://openreview.net/forum?id=x8wscCAJ2m*

## Abstract

We present a method to construct sparse neural networks using the theory of expander graphs. Expanders are sparse but well connected graph structures that are used for designing resilient networks. A Ramanujan graph is an extremal expander in terms of the spectral gap of its eigenvalues. In this work, bipartite Ramanujan expanders are deterministically constructed and used as connection structures of the convolutional and fully connected layers of a neural network. The Ramanujan graphs occur either as Cayley graphs of certain algebraic groups or as Ramanujan $r$-coverings of the full $(k, l)$ biregular bipartite graph on $k + l$ vertices. The proposed sparse networks are found to provide comparable performance to a fully dense network on benchmark datasets achieving an extremely low network density.

## 1 Introduction

Sparse neural networks are structures containing significantly lesser number of connection weights as compared to a fully connected network. They have advantages in terms of inference time, memory requirements and generalization capabilities. A sparse neural network should be trainable to achieve a high prediction accuracy. Goal of our work is to develop a theoretically justified strategy for constructing sparse neural networks that can be subsequently trained to achieve a good prediction performance.

Expander graphs are networks that are simultaneously sparse and highly connected (Hoory et al., 2006). Expanders are resilient in the sense that a large number of their edges needs to be deleted to disconnect parts of the graph. A higher spectral gap between the first and the second eigenvalues of a graph adjacency matrix points towards a better expansion property. Ramanujan graphs (Lubotzky et al., 1988) are a class of regular expander graphs with maximally high spectral gaps.

Path connectedness and regularity are desired characteristics of a neural network both for forward propagation and gradient flow. These properties help even a sparse network to achieve a good prediction performance. It has been empirically shown that the expansion property is strongly correlated with the performance of sparse neural networks (Prabhu et al., 2018; Pal et al., 2022). This motivates the use of Ramanujan graph construction algorithms to design sparse neural networks.

In our approach, bipartite Ramanujan graphs are constructed for each of the convolutional and fully connected layers. These are stacked to form the entire network. The network is then randomly initialized with weight values and trained using backpropagation. Sparsity can be controlled by varying the parameters of the Ramanujan graph. The Ramanujan graphs are constructed either as Cayley graphs of certain algebraic groups or as Ramanujan $r$-coverings of the full $(k, l)$ biregular bipartite graph on $k + l$ vertices.

Prior approaches to using Ramanujan expander graphs have relied on iterative weight pruning techniques or random graph generation. This often leads to the formation of random expander graphs. Our approach of constructing a deterministic Ramanujan network circumvents this problem. The construction is also data independent and non-iterative. Experimental results on benchmark image classification data sets show that Ramanujan sparse network initialization provides comparable performance with fully dense networks.

The paper is organized as follows. We present a brief literature survey in Section 1.1. Contributions of the paper are highlighted in Section 2. The properties and mathematical formulation of deterministic Ramanujan graphs are then presented in Section 3. The proposed construction techniques of sparse neural network layers is described in Section 4. Finally, the experimental results are outlined in Section 5.

## 1.1 Related Work

Sparse neural networks have been previously studied in terms of their graph theoretic properties (Liu et al., 2023). Connectivity properties have been used by several authors to define a sparse network topology (Vysogorets & Kempe, 2023; You et al., 2020; Chen et al., 2022; 2023a). Tam and Dunson in (Tam & Dunson, 2023) gave a technique for regularization of a neural network taking into account the connectivity property of the underlying graph. $L_0$ regularization based training which encourages zero weights has been extensively studied previously (Louizos et al., 2018). More sophisticated regularization techniques have also been used (Refael et al., 2024).

Existence of sparse trainable high performing subnetworks of a fully connected network is suggested by the well known lottery ticket hypothesis (Frankle & Carbin, 2019). These subnetworks are often identified using weight pruning techniques (Orseau et al., 2020). Initial research works in this direction were based on applying established pruning algorithms on a partially trained network (Renda et al., 2020; Fischer & Burkholz, 2022). Recently, a number of approaches has been suggested to obtain a sparse mask for pruning at initialization (Frankle et al., 2020; Pham et al., 2023; Wang et al., 2021; Sreenivasan et al., 2022; Cheng et al., 2023). Robust methods of pruning at initialization have been proposed in (Hayou et al., 2021).

Expander based sparse network generation has been studied in (Stewart et al., 2023). The methodology is based on generating random $d$-regular graphs for the bipartite layers and then filtering based on their expansion property. However, it is still a conjecture that random $d$-regular graphs are Ramanujan. A deep expander sparse network, the X-Net, is presented in (Prabhu et al., 2018). It is constructed by sampling $d$-left regular graphs from the space of all bipartite graphs. RadiX-Net (Kepner & Robinett, 2019) is a related deterministic sparse neural architecture defined over diverse layer structures. Ramanujan graph based sparse networks are proposed in (Esguerra et al., 2023). Spectral sparsification is a general method of obtaining expander like neural networks (Laenen, 2023). Expander networks used for this purpose are obtained by first generating random bipartite graphs for each layer, and then selecting the ones with a large spectral gap. However, these often favors random networks which are sensitive to reinitialization (Ma et al., 2021; Hoang et al., 2023). Spectral measures have been proposed in (Hoang et al., 2023) to prevent these effects.

## 2 Research Gap and Contributions

Most of the popular techniques for sparse network construction rely on pruning a partially trained or an untrained network. They usually consider certain heuristic weight importance functions. Expander graph networks are more theoretically justified. Existing expander graph construction methods are based on random graph generation. This does not always guarantee path-connectedness and regularity properties that are crucial for neural networks. The proposed deterministic Ramanujan graph sparse neural network construction technique circumvents these challenges.

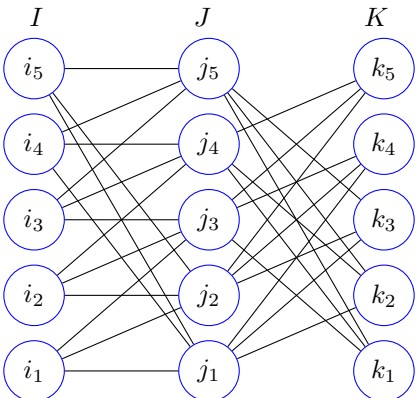

Figure 1: An example of two consecutive (edge) neural layers where each layer is a bipartite Ramanujan graph. Note that one obtains something stronger than layer-wise path-connectivity. From each vertex of (vertex) layer $I$ one has at least $d_1 d_2$ pathways to reach layer $K$ where $d_1, d_2$ denotes the regularities of layers $I, J$ respectively. For instance, to reach layer $K$ from $i_1$ there are the following 9 pathways: $i_1 \rightarrow j_1 \rightarrow k_2$, $i_1 \rightarrow j_1 \rightarrow k_3$, $i_1 \rightarrow j_1 \rightarrow k_4$, $i_1 \rightarrow j_2 \rightarrow k_3$, $i_1 \rightarrow j_2 \rightarrow k_4$, $i_1 \rightarrow j_2 \rightarrow k_5$, $i_1 \rightarrow j_3 \rightarrow k_1$, $i_1 \rightarrow j_3 \rightarrow k_4$, $i_1 \rightarrow j_3 \rightarrow k_5$.

The principal contributions of our paper are:

1. A theoretically justified methodology for generating sparse neural networks using Ramanujan graphs.

2. The method is deterministic and is guaranteed to construct well connected and regular expander graphs. It is also non-iterative and data independent.

3. We experimentally show that even extremely sparse networks can be trained to achieve comparable accuracy to a fully dense network.

## 3    Properties of the Constructed Networks

Path connectedness is a crucial property of neural network graph structures. It is necessary both for forward propagation during inference as well as for gradient flow during backpropagation (Sporns, 2003; Alford et al., 2019). Symmetry and regularity of the layer-wise adjacency matrices is another important property necessary for fast computation and memory efficiency (Chen et al., 2023b; Mao et al., 2017). We elaborate on them in the context of Ramanujan graphs.

Path-connectedness: The fact that each layer of the bipartite graphs are either regular or biregular with the regularity bigger than 3 ensures that the entire architecture remains path-connected, i.e., starting from any node in the first layer we can reach a node in the last layer by a connected path. A proof of this is direct. In Figure 1, suppose there are 3 layers $I, J, K$. We wish to reach layer $K$ starting from any point in layer $I$ by a connected path. Pick any $i_r, r \in \{1, 2, 3, 4, 5\}$. Use the fact that there is at least one edge going out from $i_r$ to reach some $j_s$ and from $j_s$ again use the fact of outgoing edges bigger than 1 to reach a point in layer $K$. The general case follows by induction on the number of layers.

High-symmetry: The adjacency matrices of Cayley graphs and of covers of Cayley graphs have much more symmetry than those of general regular graphs. For instance they are vertex transitive. Often computations are optimised to use such symmetry viz. in the case of the software GAP (Group, 2022) as an example. In the future we wish to explore this direction of using the underlying symmetry to obtain fast computations on these sparse expander networks. This is also one of the reasons why we prefer deterministic constructions over random ones as the latter loses symmetry. In the graph of Figure 1, the adjacency matrix of the first layer (which can be represented as a Cayley graph on the group $G = \mathbf{Z}_2 \times \mathbf{Z}_5{}^1 = \{(x, y) : 0 \leq x \leq 1, 0 \leq y \leq 4\}$

---

[1]$\mathbf{Z}_n$ for $n > 1$ denotes the group of integers under addition modulo $n$ and $\times$ is the Cartesian product of groups.

with generating set $S = \{(1,0), (1,1), (1,4)\}$) is

$$Adj = \begin{pmatrix} 0_{5\times 5} & B_{5\times 5} \\ B_{5\times 5}^T & 0_{5\times 5} \end{pmatrix}$$

where

$$B = \begin{pmatrix} 1 & 1 & 1 & 0 & 0 \\ 0 & 1 & 1 & 1 & 0 \\ 0 & 0 & 1 & 1 & 1 \\ 1 & 0 & 0 & 1 & 1 \\ 1 & 1 & 0 & 0 & 1 \end{pmatrix}$$

.

Our goal is to furnish an efficient sparse neural network architecture. If we can describe the architecture for each bipartite layer of neurons then we are done. The adjacency matrix of a graph captures the structure of the graph. So if we can describe the adjacency matrix or the way vertices of each bipartite layer are connected to other vertices, then it is sufficient to achieve our goal.

Previous approaches based on random network initialization and pruning strategies suffer from the issue of irregularity and are not guaranteed to be rigorously Ramanujan. For instance, application of the work of Hoory (Hoory, 2005) as mentioned in (Pal et al., 2022; Hoang et al., 2023) etc depends on the important fact that the minimal degrees of the base bipartite graphs need to be $\geq 2$ for the graphs to be Ramanujan. Our architecture based on deterministic regular Ramanujan graphs of degree $\geq 3$ ensures that the initialized networks remain Ramanujan, are path-connected and are highly symmetric, being either Cayley graphs of certain algebraic groups to replace the balanced dense bipartite graphs, or the Ramanujan $r$-covering of full biregular bipartite graphs to replace the unbalanced dense bipartite graphs.

## 4    Formulation of Sparse Neural Ramanujan Graphs

We construct a set of bipartite Ramanujan graphs of specified sparsity and then stack them as convolutional and fully connected layers of the sparse neural network. In this section we present the mathematical framework which allows us to construct these bipartite Ramanujan graphs in a deterministic manner. This forms the basis of our strategy for sparse network construction. The section proceeds as follows:

1. We shall first present the base graphs on which our sparse neural architectures will be modelled. These are the bipartite Ramanujan graphs. See Definition 4.1

2. Next we shall give the method of construction of these graphs. These graphs are used to define the neural network layers.

Recall that a Ramanujan graph is an extremal expander graph in the sense that its spectral gap is almost as large as possible. Here, we shall be concerned with bipartite Ramanujan graphs. Recall that a bipartite graph is said to be balanced if the number of vertices in each of the partitions are the same and it is said to be unbalanced otherwise.

**Definition 4.1** (Bipartite Ramanujan graphs). Let $\Gamma = (V, E)$ be a $d$-regular ($d \geq 3$) balanced bipartite graph. Let the eigenvalues of its adjacency matrix be $\lambda_n \leq \lambda_{n-1} \leq \ldots \leq \lambda_2 \leq \lambda_1$. Then $\Gamma$ is said to be Ramanujan iff $|\lambda_i| \leq 2\sqrt{d-1}$, for $i = 2, \ldots, (n-1)$.

For an unbalanced $(d_1, d_2)-$biregular bipartite graph ($d_1, d_2 \geq 3$), the condition of being Ramanujan changes to $|\lambda_i| \leq \sqrt{d_1 - 1} + \sqrt{d_2 - 1}$, for $i = 2, \ldots, (n-1)$. We see that when $d_1 = d_2$, it reduces to the usual definition. A representation of an unbalanced biregular bipartite Ramanujan network, see Figure 2. Note that we are considering undirected graphs, so the adjacency matrix is a $0-1$ symmetric matrix and the eigenvalues are all real. A bipartite graph has adjacency eigenvalues symmetric around 0. A detailed description of Ramanujan graphs can be found in (Hoory et al., 2006, sec. 5.3).

Ramanujan graphs are excellent spectral expanders. They are also extremely difficult to construct. In fact, even the question of existence of (infinite families of) Ramanujan graphs is a non-trivial one and it is not yet

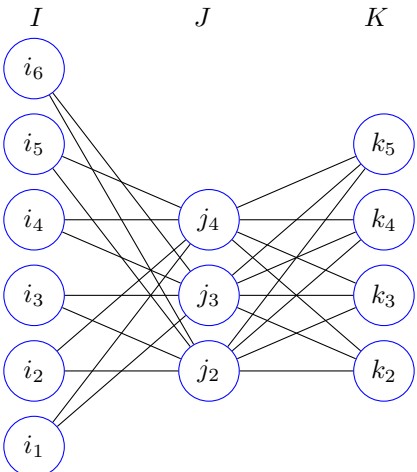

Figure 2: An example of two consecutive neural layers where each layer is a biregular bipartite Ramanujan graph. This can be checked from computing the adjacency eigenvalues of each and comparing with the Ramanujan bound for biregular bipartite graphs. The biregularity of the first layer is $(2, 4)$ and that of the second is $(4, 3)$. Note that if we have an unbalanced biregular, bipartite network with $(n_1, n_2)$ vertices and biregularity $(d_1, d_2)$ then they must be related by the following equation: $n_1 d_1 = n_2 d_2$.

fully resolved for the non-bipartite case. For the bipartite case it has been resolved by the recent works of Markus–Spielmann–Srivastava (Marcus et al., 2015; 2018) and Gribinski–Markus (Gribinski & Marcus, 2021). The first such construction of graphs are due to Lubotzky–Phillips–Sarnak (LPS) (Lubotzky et al., 1988) (and independently by Margulis (Margulis, 1988)). We shall define our pruned network according to these constructions.

## 4.1 Theoretical framework

Let us begin this section by recalling the Cheeger constant and the Cheeger inequality which will help to shed light on the fact that why good spectral expanding networks are closely related with pruned networks. In the following, a graph $\Gamma = (V, E)$ is a tuple consisting of a vertex set $V$ and an edge set $E$ which is a subset of $V \times V$.

### 4.1.1 Combinatorial Expansion

**Definition 4.2** (Expander and Cheeger constant)**.** A graph $\Gamma = (V, E)$ is an $\epsilon$-vertex expander if for every non-empty subset $X \subset V$ with $|X| \leq \frac{|V|}{2}$, we have $\frac{|\delta(X)|}{|X|} \geq \epsilon$, where $\delta(X)$ denotes the outer vertex boundary of $X$ i.e., the set of vertices in $\Gamma$ which are connected to a vertex in $X$ but do not lie in $X$. As $X$ runs over all subsets of $V$, the infimum of $\frac{|\delta(X)|}{|X|}$ satisfying the conditions above is known as the vertex Cheeger constant and is denoted by $\mathbf{h}_V(\Gamma)$.

Similar to the above, when we consider the edge boundary i.e., the set of edges which have one vertex in $X$ and the other outside of $X$, we obtain the edge Cheeger constant $\mathbf{h}_E(\Gamma)$. The vertex Cheeger constant $\mathbf{h}_V(\Gamma)$ and the edge Cheeger constant $\mathbf{h}_E(\Gamma)$ are related by the following equivalence $\frac{\mathbf{h}_V(\Gamma)}{D} \leqslant \mathbf{h}_E(\Gamma) \leqslant \mathbf{h}_V(\Gamma)$, where $D$ denotes the maximum degree of the graph. The equivalence allows us to speak about vertex expansion and edge expansion interchangeably. In the literature, having a high Cheeger constant is also known as having high combinatorial expansion. Intuitively, given a graph with high vertex (or edge) Cheeger constant, it is more difficult to separate any subset of the vertices from the rest of the graph. This allows for free flow of information throughout the network which the graph defines. Ideally, **we want our underlying graphs on which the neural layers are based to have high Cheeger constants.** In other words, we are targeting for the base graphs of the sparse neural architectures to have layerwise high Cheeger constants. However,

computing the Cheeger constants of graphs is in general an NP-hard problem. To overcome this issue we shall use spectral techniques.

### 4.1.2 Spectral Expansion

Given a finite undirected graph $\Gamma$, the eigenvalues $\lambda_n \leqslant \cdots \leqslant \lambda_1$ of its adjacency matrix are all real and $\lambda_1 \leqslant D$ with equality iff the graph is $D$-regular. (a graph is said to be $d$-regular if there are exactly $d$-edges attached to a vertex.) Thus, a $d$-regular bipartite graph is a graph which has the same number of vertices in each partition and every vertex of each partition has exactly $d$ edges attached to it. A graph $\Gamma = (V, E)$ is said to be a spectral expander if the quantities $\{|\lambda_1| - |\lambda_2|, |\lambda_1| - |\lambda_k|\}$ are both strictly positive, where $k = n - 1$ if the graph is bipartite and $k = n$ otherwise.

**Question 4.3.** *Why is there a need to establish strong spectral expansion?*

The answer to this lies in the fact that spectral expansion implies combinatorial expansion via the discrete Cheeger-Buser inequality (see appendix A.2.3) and the bigger the spectral expansion, the more combinatorially expanding the base network graphs are. In general combinatorial expansion (counting the values of the vertex or the edge Cheeger constants) is an NP-hard problem. Now the natural question arises that whether there is a limit to the spectral expansion or can it become as large as possible. This leads us to Ramanujan graphs which are the optimal spectral expanders. See appendix for the Alon-Bopanna theorem, Thm A.3.

Now, we proceed to the construction of the deterministic Ramanujan networks. We shall need the following notions from arithmetic.

**Definition 4.4** (Quadratic residue and Legendre symbol)**.** An integer $q$ is called a quadratic residue modulo $n$ if it is congruent to a perfect square modulo $n$, i.e., if there exists an integer $x$ such that $x^2 \equiv q \pmod{n}$. Otherwise, $q$ is called a quadratic non-residue modulo $n$.
Let $p$ be an odd prime number and $a$ be an integer. The Legendre symbol of $a$ and $p$ is defined as

$$\left(\frac{a}{p}\right) = \begin{cases} 1 & \text{if } a \text{ is a quadratic residue modulo } p \\ & \quad \text{and } a \not\equiv 0 \pmod{p}, \\ -1 & \text{if } a \text{ is a quadratic non-residue modulo } p, \\ 0 & \text{if } a \equiv 0 \pmod{p}. \end{cases}$$

Given a prime $a$, there are infinitely many primes $p$ such that Legendre symbol of $a$ and $p$ is $-1$ (and also there are infinite many primes $p$ such that it is $+1$).
Motivation: The bipartite Ramanujan networks will be constructed as Cayley graphs of certain matrix groups over finite fields. In other words, the vertices will be matrices while the coefficients of the matrices will be elements of certain finite fields. How each matrix element is connected to the other matrix elements (essentially the same thing as saying how each vertex in the graph network is connected to any other vertex) will depend on the Quadratic residue and the Legendre symbols defined above. We shall now see which matrix groups we will be working with.

**Definition 4.5** ($PGL_2(\mathbb{K})$ projective linear group over a field $\mathbb{K}$)**.** Let $\mathbb{K}$ be any field. Let us denote by $GL_2(\mathbb{K})$ the group of invertible 2-by-2 matrices with coefficients in $\mathbb{K}$, ie, the matrices with non-zero determinant. Let $PGL_2(\mathbb{K})$ be the quotient group

$$PGL_2(\mathbb{K}) = GL_2(\mathbb{K})/Z(\mathbb{K})$$

where

$$GL_2(\mathbb{K}) = \left\{ \begin{pmatrix} a & b \\ c & d \end{pmatrix} : ad - bc \neq 0 \right\}$$

and

$$Z(\mathbb{K}) = \left\{ \begin{pmatrix} a & 0 \\ 0 & a \end{pmatrix} : a \neq 0 \right\}$$

In these definitions it is assumed that all entries are elements of the field $\mathbb{K}$.
Remark: If $\mathbb{K} = \mathbb{F}_q$ (the finite field of $q$ elements), then a simple computation gives the size of $PGL_2(\mathbb{F}_q)$

to be $|PGL_2(\mathbb{F}_q)| = q(q^2 - 1)$. In Section 4.3, we shall use this property to construct bipartite $\frac{q(q^2-1)}{2}$ by $\frac{q(q^2-1)}{2}$ Ramanujan networks. For more details see appendix, A.3.

## 4.2 Regular Ramanujan graphs

Let $p, q \equiv 1 (\text{mod } 4)$ be distinct odd primes (the condition of $\equiv 1 (\text{mod } 4)$ can be removed at the cost of making the analysis more technical and complicated, we shall mention later how it is achieved). The graph $X^{p,q}$ is constructed using the following general method ((Lubotzky et al., 1988)).

1. Choice of group: It will be a Cayley graph (for the notion of Cayley graphs, see Appendix A.1) on $PGL_2(\mathbb{F}_q)$. Thus the vertices of the graph will be elements of $PGL_2(\mathbb{F}_q)$. Next, we need to take care of the edges.

2. Method of selecting certain special matrices within $PGL_2(\mathbb{F}_q)$: Consider the equation $a_0^2 + a_1^2 + a_2^2 + a_3^2 = p$. Jacobi's four square theorem states that there are $p+1$ solutions to the equation $a_0^2 + a_1^2 + a_2^2 + a_3^2 = p$ with $a_0 > 0$ odd (i.e., $a_0 \equiv 1 \pmod{2}$) and $a_1, a_2, a_3$ even. The role of the finite field $\mathbb{F}_q$ will now come into play. We have a set of solutions $\{(a_0, a_1, a_2, a_3)\}$ (in the integers) of size $p + 1$ to the equation $a_0^2 + a_1^2 + a_2^2 + a_3^2 = p$. For each such solution $(a_0, a_1, a_2, a_3)$ consider the matrix $\begin{pmatrix} a_0 + ia_1 & a_2 + ia_3 \\ -a_2 + ia_3 & a_0 - ia_1 \end{pmatrix}$ where $i$ is some fixed integer solution to $i^2 = -1 \pmod{q}$. We will obtain a set of $(p + 1)$ matrices each of which belongs to $PGL_2(\mathbb{F}_q)$ (can be checked from the definition of $PGL_2(\mathbb{F}_q)$).

   For example, suppose we take $q = 5$, $p = 13$, and $i = 3$ (where $i^2 \equiv -1 \mod 5$).

   Let us choose the solution:
   $$a_0 = 3, \quad a_1 = 0, \quad a_2 = 0, \quad a_3 = 2,$$
   to the equation $a_0^2 + a_1^2 + a_2^2 + a_3^2 = 13$.

   Now, we construct the matrix:
   $$\begin{pmatrix} a_0 + ia_1 & a_2 + ia_3 \\ -a_2 + ia_3 & a_0 - ia_1 \end{pmatrix} = \begin{pmatrix} 3 + 3 \cdot 0 & 0 + 3 \cdot 2 \\ -0 + 3 \cdot 2 & 3 - 3 \cdot 0 \end{pmatrix} = \begin{pmatrix} 3 & 6 \\ 6 & 3 \end{pmatrix}.$$

   Next, we reduce the entries modulo 5 (since $q = 5$):
   $$\begin{pmatrix} 3 & 6 \mod 5 \\ 6 \mod 5 & 3 \end{pmatrix} = \begin{pmatrix} 3 & 1 \\ 1 & 3 \end{pmatrix}.$$

   Thus, an example of such a chosen matrix is:
   $$\begin{pmatrix} 3 & 1 \\ 1 & 3 \end{pmatrix}.$$

   This matrix is one of the $p + 1 = 14$ matrices generated from the solutions to the equation $a_0^2 + a_1^2 + a_2^2 + a_3^2 = 13$ using the process described.

3. Form the generating set $S$ of the Cayley graph (see Appendix A.1) to be the set of the $(p + 1)$ matrices created above. The vertices are the elements of $PGL_2(\mathbb{F}_q)$ and the edges are given $\{(x, xs) : x \in PGL_2(\mathbb{F}_q), s \in S\}$. In other words, $X^{p,q} = Cay(PGL_2(\mathbb{F}_q), S)$ cf. A.1.

The graphs are bipartite iff $p$ is not a quadratic residue modulo $q$ or in other words the Legendre symbol $\left(\frac{q}{p}\right) = -1$. The bipartite graphs $X^{p,q}$ will be $(p+1)$-regular, of size $\frac{q(q^2-1)}{2}$ by $\frac{q(q^2-1)}{2}$ and are Ramanujan. For a proof see (Lubotzky et al., 1988).

Remark: If $p \equiv 3 \pmod{4}$, then a similar strategy is employed, except in this case one looks at solutions of $a_0^2 + a_1^2 + a_2^2 + a_3^2 = p$ with $a_0 \equiv 0 \pmod{2}$. See (Musitelli & de la Harpe, 2006, sec. 2).

### 4.3 Construction for the fully connected layers

For the fully connected layers consisting of balanced bipartite graphs of size $m \times m$ and regularity $m$ where $m$ is the number of vertices on each side of the graph, we prune them at initialization in accordance with the Ramanujan graph structure of Lubotzky–Phillips–Sarnak (Lubotzky et al., 1988).

1. Select the largest value of prime $q \equiv 1 \pmod 4$ such that $\frac{q(q^2-1)}{2} \leq m$.

2. Select a prime $p \equiv 1 \pmod 4$ such that the Legendre symbol (4.4) $\left(\frac{q}{p}\right) = -1$ (note that this choice is always possible as given a prime $q$ there are infinite number of primes $p$ satisfying this property)

3. Follow the construction using these values of $p$ and $q$ as given in Section 4.2.

Selecting the minimum possible value of $p$ will give us the sparsest possible Ramanujan graph. For example, consider a $4096 \times 4096$ original fully connected layer, our choice of $(p, q) = (5, 17)$ gives rise to bipartite sparse Ramanujan graphs of size $2448 \times 2448$ with regularity 6.

Note: Here we have taken $p \equiv 1 \pmod 4$, but we could have also chosen $p \equiv 3 \pmod 4$ or even $p \equiv 2 \pmod 4$ (see construction of cubic Ramanujan graphs (Chiu, 1992)) resulting in even sparser networks.

### 4.4 biregular Ramanujan graphs

A bipartite graph is said to be $(d_1, d_2)$ biregular if each bi-partition has fixed regularity $d_1$, $d_2$ respectively. Note that a simple computation reveals that if $(n_1, n_2)$ are the bi-partition sizes, then $n_1 d_1 = n_2 d_2$. Thus three parameters are needed to specify these types of graphs. One way to construct biregular Ramanujan graphs is the following, see (Burnwal et al., 2021):

1. Select a prime $q$ and create a $q \times q$ cyclic shift permutation matrix $P = [P]_{ij}$ with $[P]_{ij} = 1$ if $j = i - 1 \pmod q$ and 0 otherwise.

2. Select any value of $l \leq q$ and create the bi-adjacency matrix $B = \begin{pmatrix} I_q & I_q & \dots & I_q \\ I_q & P & \dots & P^{l-1} \\ I_q & P^2 & \dots & P^{2(l-1)} \\ \vdots & & & \\ I_q & P^{q-1} & \dots & P^{(q-1)(l-1)} \end{pmatrix}$

   where $I_q$ is the $q \times q$ identity matrix and $P$ is as above.

3. The biadjacency matrix $B$ can be used to generate adjacency matrix of any $m \times n$ bipartite graph as $Adj = \begin{pmatrix} 0_{m \times m} & B_{m \times n} \\ B_{m \times n}^T & 0_{n \times n} \end{pmatrix}$.

$B$ is a $q^2 \times lq$ matrix and the bipartite graph is of size $q^2 \times lq$ with biregularity $(l, q)$. The graphs whose bi-adjacency matrices are represented as $B$ are Ramanujan. These graphs are explicit realisations of the Ramanujan $r$-coverings of the full $(k, l)$ biregular bipartite graph on $k + l$ vertices as shown in (Hall et al., 2018, cor 2.2).

Note: The condition $q \geq l$ is not a necessary requirement. The reason behind this flexibility lies in the specific properties of the bi-adjacency matrix $B$, which has dimensions $q^2$ by $lq$. The critical insight is that if $B$ creates a Ramanujan graph, then its transpose, denoted as $B^T$ (with dimensions $lq$ by $q^2$), also creates a Ramanujan graph.

### 4.5 Construction for the convolutional layers

The convolutional layer can be thought of as a matrix of dimensions $|N_{out}| \times |N_{in}| \times |K_w| \times |K_h|$ where $|N_{out}|$ is the number of output channels, $|N_{in}|$ is the number of input channels, $|K_w|$ is the kernel width and

$|K_h|$ is the kernel height. This is considered to be a bipartite graph with $|V_{left}| = |N_{in}| \times |K_w| \times |K_h|$ and $V_{right} = |N_{out}|$ where each vertex of $V_{left}$ has an edge with each vertex of $V_{right}$.

1. Select the largest value of prime $q$ such that $q^2 \leq |V_{left}|$.

2. Select the largest value of $l$ such that $l \leq q$ and $lq \leq |V_{right}|$.

3. Follow the construction as given in Section 4.4.

The pruning mask thus obtained is a matrix of size $q^2 \times lq$ with the effective number of connections being equal to $q^2 \times l$. The original pruning mask of the convolutional layer has size $|N_{out}| \times [|N_{in}| \times |K_w| \times |K_h|]$. By construction the obtained Ramanujan graph is actually a subgraph of the original pruning mask and thus the entries in the original mask not part of the constructed Ramanujan graph are set to 0.

Note: The above construction assumes $V_{left} \geq V_{right}$. If this is not the case, we can simply swap the roles of $q$ and $l$ since from the construction (4.4), we know $B^T$ also creates Ramanujan graphs.

### 4.6 Time Complexity of Construction

The time complexity for constructing ramanujan graphs for fully connected layers following Section 4.3 is mainly dominated by the creation of the $PGL_2$ group in which first we need to create the generator matrix and then find the equivalence classes which takes time $O(q^4 \times q)$. The solution to the four square problem has complexity $O(p^4)$ giving an overall complexity of $O(q^5 + p^4)$. Similarly for constructing Ramanujan graphs for convolutional layers as given in Section 4.5 the time complexity comes out to be $O(q^2 \times lq)$ since we need to create the bi-adjacency matrix $B$.

### 4.7 General bipartite networks

In the case of bipartite networks with arbitrary sizes, one can achieve as sparse Ramanujan graphs as possible. It has recently been proven by Marcus, Spielman and Srivastava that for the regular case, for each degree $d \geq 3$, infinite families of bipartite Ramanujan graphs exist. This is also true for the biregular case, for each pair $(d_1, d_2)$ with $d_1, d_2 \geq 3, d_2 = kd_1, k \geq 2$. Further, they showed the existence of these types of graphs of all sizes. Their method of proof is probabilistic and existential in nature. It does not give explicit families of bipartite Ramanujan graphs. However there now exist polynomial time algorithms (Cohen, 2016) (for the regular case) (Gribinski & Marcus, 2021) (for the biregular case) with which we can extract explicit Ramanujan graphs. For the regular case, we fix an integer $n \geq 3$ and a degree $d \geq 3$ and in the output we shall obtain a $d$-regular $n \times n$ bipartite Ramanujan graph while for the biregular case, we fix three integers $n, k, d$ with $n > 2, d > 2, k \geq 2$ and obtain $(d, kd)$ biregular Ramanujan graph of size $kn \times n$.

## 5 Experimental Methodology and Results

The goal of our experiments is to study the effectiveness of deterministic Ramanujan graph based sparse network construction.

### 5.1 Datasets and architectures

The datasets used for the experiments are CIFAR-10 and CIFAR-100 (Krizhevsky, 2009). The experiments are performed over a variety of architectures including VGG13, VGG16, VGG19 (Simonyan & Zisserman, 2014), AlexNet (Krizhevsky et al., 2012), ResNet18 and ResNet34 (He et al., 2016) to show the robustness of our method. We proceed in two parts. In the first part, we prune the Fully Connected layers by replacing them with sparse Ramanujan Graph which is applicable for VGG13, VGG19 and AlexNet architectures. In the second part, we prune the whole network including the Convolutional layers and the Fully Connected layers which is applicable for all the architectures considered in our experiment. The performance of the fully dense and the pruned networks are compared in each case. Finally, we compare the performance of our method against various sparse neural networks obtained by state-of-the art pruning at initialisation algorithms

for VGG16 and the ResNet34 architectures. Training parameters for all of the architectures are same and are summarized in Table 1. We report accuracy on a randomly split 16% test set for all the experiments.

Table 1: Training Parameters for the experiment

| Hyperparmeters | |
| --- | --- |
| Epochs | 200 |
| Train Batch Size | 256 |
| Test Batch Size | 128 |
| Learning Rate | 0.1 |
| LR Decay, Epoch | 10x, [100, 150] |
| Optimizer | SGD |
| Weight Decay | 0.0005 |
| Momentum | 0.9 |
| Weight Initialization | Kaiming Uniform |

## 5.2 Methods compared

The performance of the pruned networks is compared with that of a fully dense network having similar number of nodes in each layer. We also compare our method with other sparse neural networks obtained by state-of-art pruning at initialization techniques. Like our approach, these methods generate sparse connectivity structures which are initialized to random weight values and then trained using backpropagation. The pruning methods compared include, Random (Liu et al., 2022), ERK (Evci et al., 2020; Mocanu et al., 2018), GraSP (Wang et al., 2020), and SynFlow (Tanaka et al., 2020). The number of iterations used for SynFlow are 100 and for ERK and GraSP it is 1 keeping the rest of the hyperparameters same as in Table 1.

## 5.3 Network construction parameters

Experiments are conducted in two parts: 1) Considering sparse Fully Connected layers only, 2) Considering the entire network to be sparse including the Convolutional and the Fully Connected layers. For the first part, we consider sparse fully connected layers using the construction of Ramanujan graphs as given in Section 4.3. We have used a dedicated fully connected layer of size $4096 \times 4096$ for VGG13, VGG19 and AlexNet architectures and the values used for $p$ and $q$ (Section 4.3) are given in Table 2. This results in the fully connected layer becoming of size $q(q^2 - 1)/2 \times q(q^2 - 1)/2$ with the effective number of connections between the layer being equal to $q \times (q^2 - 1)/2 \times (p + 1)$.

Table 2: Values of $p$ and $q$ used to generate the sparse fully connected layer

| Model | VGG13, VGG19, AlexNet |
| --- | --- |
| FC Layer Size | $4096 \times 4096$ |
| $p$ | 29, 109 |
| $q$ | 17 |

For the second part, we consider sparse convolutional layers using the construction of Ramanujan graphs as given in Section 4.5 in addition to the fully connected layers. The size of convolutional layer being pruned and the values of $l$ and $q$ used (Section 4.5) for VGG16 and ResNet34 architectures is given in Table 3, while for the rest of the architectures is given in Table 7.

The network density (ratio of number of edges in a sparse network to a fully dense network) reported in Section 5.4 is calculated by dividing the number of connections which is equal to the sum of $q \times (q^2 - 1)/2 \times (p + 1)$ (number of connections in fully connected layer) and $q^2 \times l$ (number of connections in each of the convolutional layers) divided by the total number of connections present in the fully dense network with the similar number of nodes.

Table 3: Values of $q$ and $l$ to generate Ramanujan Graphs for layers of VGG16 and ResNet34

| VGG16 | | | ResNet34 | | |
|---|---|---|---|---|---|
| convolutional layer Size | q | l | convolutional layer Size | q | l |
| $[128 \times 64 \times 3 \times 3] \times 1$ | 11 | 52 | $[64 \times 64 \times 3 \times 3] \times 6$ | 7 | 82 |
| $[128 \times 128 \times 3 \times 3] \times 1$ | 11 | 104 | $[128 \times 64 \times 3 \times 3] \times 1$ | 11 | 52 |
| $[256 \times 128 \times 3 \times 3] \times 1$ | 13 | 88 | $[128 \times 128 \times 3 \times 3] \times 7$ | 11 | 104 |
| $[256 \times 256 \times 3\times] \times 2$ | 13 | 177 | $[256 \times 128 \times 3 \times 3] \times 1$ | 13 | 88 |
| $[512 \times 256 \times 3 \times 3] \times 1$ | 19 | 121 | $[256 \times 256 \times 3 \times 3] \times 11$ | 13 | 177 |
| $[512 \times 512 \times 3 \times 3] \times 5$ | 19 | 242 | $[512 \times 256 \times 3 \times 3] \times 1$ | 19 | 121 |
| | | | $[512 \times 512 \times 3\times] \times 5$ | 19 | 242 |

## 5.4 Results and discussion

We study the accuracy of sparse networks obtained by our technique for various architectures and datasets. The accuracy is compared with that of a fully dense network with similar number of nodes. Results for the first part of the experiment where only the intermediate fully connected layer is sparse, are summarized in Table 4. By definition, network density is the ratio of number of edges in the sparse network to the number of edges in a fuly dense network. It can be observed that the Ramanujan graph construction allows us to extremely sparsify the fully connected layer to a network density of **0.43%** while still having comparable the accuracy as of the similar fully dense model.

Table 4: Accuracy of VGG and AlexNet when only the FC layer is sparsified

| Dataset: CIFAR-10 | | | | Dataset: CIFAR-100 | | | |
|---|---|---|---|---|---|---|---|
| Model | FC layer Size (Network Density) | | | Model | FC layer Size (Network Density) | | |
| | $4096 \times 4096$ (Unpruned) | $2448 \times 110$ (**1.6%**) | $2448 \times 30$ (**0.43%**) | | $4096 \times 4096$ (Unpruned) | $2448 \times 110$ (**1.6%**) | $2448 \times 30$ (**0.43%**) |
| VGG13 | 92% | **91%** | **91%** | VGG13 | 66% | **66%** | **63%** |
| VGG19 | 92% | **92%** | **92%** | VGG19 | 66% | **67%** | **63%** |
| AlexNet | 86% | **84%** | **86%** | AlexNet | 67% | **66%** | **66%** |

For the second part of the experiment where we sparsify the complete network including the convolutional layers and the fully connected layer, we could achieve an overall network density of ∼2% to ∼5% for VGG, ∼2.3% for AlexNet and ∼5% for the ResNet architectures. The accuracy of the models on the CIFAR-10 and CIFAR-100 datasets are summarized in Table 5. A small accuracy drop is observed as compared to the fully dense network.

Table 5: Accuracy of various architectures when the complete network is sparsified including the Convolutional and the FC layers

| Dataset: CIFAR-10 | | | |
|---|---|---|---|
| Model | Unpruned accuracy | Pruned Accuracy | Network Density |
| VGG13 | 92% | 90% | 1.7% |
| VGG16 | 93% | 91% | 5.3% |
| VGG19 | 92% | 89% | 2.4% |
| AlexNet | 86% | 82% | 2.3% |
| ResNet18 | 87% | 86% | 5.6% |
| ResNet34 | 88% | 86% | 5.2% |

| Dataset: CIFAR-100 | | | |
|---|---|---|---|
| Model | Unpruned accuracy | Pruned Accuracy | Network Density |
| VGG16 | 70% | 66% | 5.3% |
| ResNet18 | 55% | 54% | 5.6% |
| ResNet34 | 57% | 56% | 5.2% |

| Dataset: Tiny-ImageNet | | | |
|---|---|---|---|
| Model | Unpruned accuracy | Pruned Accuracy | Network Density |
| VGG16 | 43% | 40% | 5.3% |
| ResNet34 | 56% | 48% | 5.2% |

Finally, we compare the performance of the proposed Ramanujan sparse network initialization with sparse networks obtained by state-of-art pruning at initialization techniques. The comparison of accuracy between various pruning at initialization techniques at network density ∼5% is shown for the VGG16 and ResNet34 architectures in Table 6.

Table 6: Comparison of our method against sparse neural networks obtained by pruning at initialization

| Dataset: CIFAR-10 | | Dataset: CIFAR-100 | | Dataset: Tiny-ImageNet | |
|---|---|---|---|---|---|
| VGG16 (Network Density ∼5.3%) | | VGG16 (Network Density ∼5.3%) | | VGG16 (Network Density ∼5.3%) | |
| Method | Accuracy | Method | Accuracy | Method | Accuracy |
| Unpruned | 93% | Unpruned | 70% | Unpruned | 43% |
| Our Method | 91% | **Our Method** | **66%** | **Our Method** | **40%** |
| Random | 89% | Random | 60% | Random | 35% |
| ERK | 91% | ERK | 62% | ERK | 39% |
| **SynFlow** | **92%** | SynFlow | 65% | **SynFlow** | **40%** |
| ResNet34 (Network Density ∼5.2%) | | ResNet34 (Network Density ∼5.2%) | | ResNet34 (Network Density ∼5.2%) | |
| Method | Accuracy | Method | Accuracy | Method | Accuracy |
| Unpruned | 88% | Unpruned | 57% | Unpruned | 56% |
| **Our Method** | **86%** | **Our Method** | **56%** | **Our Method** | **48%** |
| Random | 81% | Random | 50% | Random | 45% |
| **ERK** | **86%** | **ERK** | **56%** | ERK | 47% |
| **GraSP** | **86%** | **GraSP** | **56%** | GraSP | 29% |

We can observe that the proposed sparse neural network can achieve comparable accuracy to other sparse networks obtained by iterative pruning at initialization techniques. It also significantly outperforms the randomly connected sparse network. The accuracy of the Ramanujan graph based sparse networks remains close to that of a similar fully dense network even at a low network density.

## 6 Conclusion and Future Work

We presented a technique for constructing sparse neural networks that can be trained to a high accuracy. The method is based on a deterministic Ramanujan graph construction technique using Cayley graphs and Ramanujan coverings. The construction is used for generating sparse and regular bipartite graphs which are stacked to obtain fully connected and convolutional layers for CNN and ResNets. Experimental results on benchmark data sets demonstrate that even a very sparse Ramanujan network structure can achieve comparable accuracy to that of a fully dense network with similar number of layers and nodes.

With the success of sparse deterministic Ramanujan neural networks, our further direction of work is to implement these in the case of transformers and study sparse Ramanujan transformer networks. The proposed deterministic construction technique is expected to significantly reduce the number of parameters and training time while maintaining high accuracy.

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

# A  Appendix

## A.1  Cayley graph

Let $G$ be a group and let $S$ be a subset of $G$ that is closed under inversion i.e., $S = S^{-1}$. The corresponding Cayley graph $C(G, S)$ is a graph with vertex set the elements of $G$ and edge set $\{(x, xs) : x \in G, s \in S\}$. As an example of a Cayley graph of a non-abelian group, one can take the group $G = D_4$, the dihedral group of order 8 with elements $\{e, r, r^2, r^3, s, sr, sr^2, sr^3\}$ and generating set $S = \{r, s, r^{-1}\}$. Here the $r$ denotes rotation by $\frac{\pi}{2}$ and $s$ is reflection. So $r^4 = 1$, $s^2 = 1$ and $sr = r^{-1}s$.

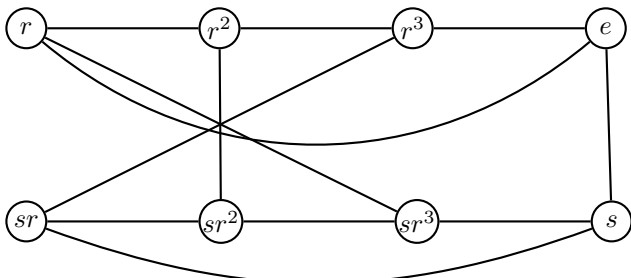

The Cayley Graph $C(D_4, \{s, r, r^3\})$

## A.2 Expander graphs and Ramanujan graphs

An expander graph is a structurally sparse graph that has strong connectivity properties. The connectivity can be quantified in different ways which give rise to different notions of expanders such as vertex expanders, edge expanders and spectral expanders. These notions are actually interrelated. In the following, a graph $\Gamma = (V, E)$ is a tuple consisting of a vertex set $V$ and an edge set $E$ which is a subset of $V \times V$.

### A.2.1 Combinatorial expansion

**Definition A.1** (vertex Cheeger constant). The infimum of the quantity $\frac{|\delta(X)|}{|X|}$ where $\delta(X)$ denotes the outer vertex boundary of $X$ i.e., the set of vertices in $\Gamma$ which are connected to a vertex in $X$ but do not lie in $X$ as $X$ runs over all non-empty subsets of $V$ satisfying the condition with $|X| \leq \frac{|V|}{2}$ is known as the vertex Cheeger constant and is denoted by $\mathbf{h}_V(\Gamma)$.

**Definition A.2** (edge-Cheeger constant). The edge boundary of a set $S$, denoted $\delta S$, is $\delta S =$ the set of edges going out from $S$ to its complement. The edge Cheeger constant of $\Gamma$, denoted by $\mathbf{h}_E(\Gamma)$, is defined as: $\mathbf{h}_E(\Gamma) = \min \frac{|\delta S|}{D|S|}$ as $S$ satisfies the following: $\{S \neq \text{empty set}, |S| \leq \frac{n}{2}\}$ and $D$ is the maximum degree of the graph $\Gamma$

The vertex Cheeger constant $\mathbf{h}_V(\Gamma)$ and the edge Cheeger constant $\mathbf{h}_E(\Gamma)$ are related by the following equivalence

$$\frac{\mathbf{h}_V(\Gamma)}{D} \leq \mathbf{h}_E(\Gamma) \leq \mathbf{h}_V(\Gamma),$$

where $D$ denotes the maximum degree of the graph (the degree of each vertex is the number of edges going out from the vertex). This allows one to speak about vertex expansion and edge expansion interchangeably. Having high combinatorial expansion means having high Cheeger constant, a desirable property for our case.

### A.2.2 Spectral expansion

Given a finite undirected graph $\Gamma$ the eigenvalues $\lambda_n \leq \cdots \leq \lambda_1$ of its adjacency matrix are all real and $\lambda_1 \leq D$ with equality iff the graph is $D$-regular. The spectra, i.e., the distribution of the eigenvalues convey a lot of information about the structure of the graphs. For instance, the quantity $\lambda_1 - \lambda_2$ (also known in the literature as the one sided spectral gap) quantifies the connectivity and the combinatorial expansion of the graph via the discrete Cheeger-Buser inequality, discovered independently by (Dodziuk, 1984) and by (Alon & Milman, 1985). A graph $\Gamma = (V, E)$ is said to be a spectral expander if the quantities $\{|\lambda_1| - |\lambda_2|, |\lambda_1| - |\lambda_k|\}$ are both bounded away from zero, where $k = n - 1$ if the graph is bipartite and $k = n$ otherwise.

### A.2.3 Discrete Cheeger–Buser inequality

The discrete Cheeger–Buser inequality discovered independently by (Dodziuk, 1984) and by (Alon & Milman, 1985) allows one to pass from spectral expansion to combinatorial expansion. The inequality states that

$$\frac{\mathbf{h}_E(\Gamma)^2}{2} \leq \alpha_2 \leq 2\mathbf{h}_E(\Gamma),$$

where $\alpha_2$ denotes the second smallest eigenvalue of the normalised Laplacian matrix of $\Gamma$ and is related to the eigenvalues of the adjacency matrix via

$$\frac{\lambda_i}{D} \leqslant 1 - \alpha_i \leqslant \frac{\lambda_i}{d} \ \forall i = 1, 2, \dots, n,$$

where $D$ and $d$ denote the maximal and minimal degrees respectively. See (Chung, 2016) for details. From the above, it is easy to check that a high $|\lambda_1| - |\lambda_2|$ ensures a high $\mathbf{h}_E(\Gamma)$ and vice-versa. Thus, the two notions of expansion are inter-connected and every spectral expander remains a combinatorial expander. They are actually equivalent for some classes of graphs, for instance bipartite graphs (as the adjacency spectrum is symmetric about the origin), variants of algebraic graphs e.g., see (Breuillard et al., 2015; Biswas, 2019; Biswas & Saha, 2021; 2023) etc.

### A.2.4 Ramanujan graph bounds, Alon-Bopanna Theorem

A $d$-regular graph is said to be a Ramanujan graph if $\max\{|\lambda_2|, |\lambda_k|\} \leqslant 2\sqrt{d-1}$. In the case of bipartite graphs, $\lambda_n = \lambda_1$ and $\lambda_{n-1} = \lambda_2$, hence the previous expression reduces to $|\lambda_2| \leqslant 2\sqrt{d-1}$. For fixed degree, with the sizes of the graphs growing larger and larger, these are the best possible expanders, as given by the Alon-Bopanna bound (Alon, 1986; Nilli, 1991).

**Theorem A.3** (Alon-Boppana). *For every $d$ regular graph on $n$ vertices,*

$$\lambda \geq 2\sqrt{d-1} - o_n(1).$$

*The $o_n(1)$ term is a quantity that tends to zero for every fixed $d$ as $n \to \infty$.*

The bound requires a bit of technical details. However, if one relaxes a bit the right hand side of the above bound then one can easily show the following,

$$\lambda \geq \sqrt{d} \cdot (1 - o_n(1)).$$

The proof goes as follows: Let $A$ be the adjacency matrix of $G$, then $\mathrm{trace}(A^k)$ is the number of all walks of length $k$ in $G$ that start and end in the same vertex. In particular, all the diagonal entries in $A^2$ are $\geq d$. Thus, $\mathrm{trace}(A^2) \geq nd$. On the other hand,

$$\mathrm{trace}(A^2) = \sum_i \lambda_i^2 \leq d^2 + (n-1)\lambda^2.$$

Thus, $(n-1)\lambda^2 \geq dn - d^2$, which implies that $\lambda^2 \geq d \cdot \frac{n-d}{n-1}$.
See (Hoory et al., 2006) for details.

### A.2.5 Expanders and Ramanujan graphs from finite simple groups

The existence and construction of expanders are a deep question and those of Ramanujan graphs are even deeper. Most of the constructions of expanders are based on Cayley graphs of finite simple groups of Lie type. The first construction is due to Margulis (Margulis, 1982). Later Lubotzky–Phillips–Sarnak (Lubotzky et al., 1988) constructed Ramanujan graphs from $SL_2(\mathbb{F}_p)$. Till 2014 these graphs and variants thereof by (Chiu, 1992) and (Morgenstern, 1994) were the only known construction of Ramanujan graphs. Recent works of Markus–Spielmann–Srivastava (Marcus et al., 2015) have shown the existence of bipartite Ramanujan graphs of all degrees and sizes. Recently, there has also been new research directions on the topic of construction of expanders satisfying other desirable properties such as the diameter-by-girth ratio are bounded, for instance (Arzhantseva & Biswas, 2022). It will be interesting to see if these special expander networks play important roles as architectures for neural networks or not.

**Question A.4.** *Why are the LPS graphs (Lubotzky et al., 1988) (sec 4.2) and the graphs presented in sec 4.4 Ramanujan?*

For the LPS graphs, the proof is quite involved, a nice exposition on the proof can be found in (Davidoff et al., 2003), Chapter 4 and also in (Sarnak, 1990) Chapter 3. For the graphs presented in sec 4.4, a direct computation of the eigenvalues using the propoerties of cyclic shift permutation matrices is enough. See proof of Theorem 3 in (Burnwal et al., 2021).

### A.3 Projective Linear Groups Over Finite Fields

The projective linear group $PGL_n(\mathbb{F}_q)$ is a matrix group defined as the group of linear transformations on the projective space over $\mathbb{F}_q$, modulo scalar transformations.

- Let $\mathbb{F}_q$ be a finite field with $q$ elements.

- Let $GL_n(\mathbb{F}_q)$ denote the general linear group of $n \times n$ invertible matrices over $\mathbb{F}_q$.

- The projective linear group $PGL_n(\mathbb{F}_q)$ is defined as:

$$PGL_n(\mathbb{F}_q) = \frac{GL_n(\mathbb{F}_q)}{Z(GL_n(\mathbb{F}_q))}$$

where $Z(GL_n(\mathbb{F}_q))$ is the center of $GL_n(\mathbb{F}_q)$, which consists of scalar matrices.

**Matrix Form of $PGL_2(\mathbb{F}_q)$**

The projective general linear group $PGL_2(\mathbb{F}_q)$ is the group of projective transformations of the projective line over $\mathbb{F}_q$. It can be expressed in terms of matrices as follows:

The general linear group $GL_2(\mathbb{F}_q)$ consists of all invertible $2 \times 2$ matrices over $\mathbb{F}_q$. A matrix $A \in GL_2(\mathbb{F}_q)$ is given by:

$$A = \begin{pmatrix} a & b \\ c & d \end{pmatrix}$$

where $a, b, c, d \in \mathbb{F}_q$ and $\det(A) = ad - bc \neq 0$.

**Center of $GL_2(\mathbb{F}_q)$**

The center of $GL_2(\mathbb{F}_q)$ consists of scalar matrices of the form:

$$Z(GL_2(\mathbb{F}_q)) = \left\{ \lambda I \mid \lambda \in \mathbb{F}_q^* \right\}$$

where $I$ is the identity matrix:

$$\lambda I = \begin{pmatrix} \lambda & 0 \\ 0 & \lambda \end{pmatrix}$$

and $\mathbb{F}_q^*$ is the multiplicative group of the field $\mathbb{F}_q$.

The projective general linear group $PGL_2(\mathbb{F}_q)$ is defined as the quotient of $GL_2(\mathbb{F}_q)$ by its center:

$$PGL_2(\mathbb{F}_q) = \frac{GL_2(\mathbb{F}_q)}{Z(GL_2(\mathbb{F}_q))}$$

In matrix terms, an element of $PGL_2(\mathbb{F}_q)$ can be represented by a matrix $A \in GL_2(\mathbb{F}_q)$, where matrices are considered equivalent if they differ by a scalar multiple. Thus, the matrix form of $PGL_2(\mathbb{F}_q)$ is represented by equivalence classes of matrices of the form:

$$\left\{ \begin{pmatrix} a & b \\ c & d \end{pmatrix} \mid a, b, c, d \in \mathbb{F}_q \text{ and } ad - bc \neq 0 \right\}$$

modulo the scalar matrices.

The distinct elements of $PGL_2(\mathbb{F}_2)$ are represented by:

$$\left\{ \begin{pmatrix} 1 & 0 \\ 0 & 1 \end{pmatrix}, \begin{pmatrix} 1 & 0 \\ 1 & 1 \end{pmatrix}, \begin{pmatrix} 1 & 1 \\ 0 & 1 \end{pmatrix}, \begin{pmatrix} 0 & 1 \\ 1 & 0 \end{pmatrix}, \begin{pmatrix} 0 & 1 \\ 1 & 1 \end{pmatrix}, \begin{pmatrix} 1 & 1 \\ 1 & 0 \end{pmatrix} \right\}$$

$PGL_2(\mathbb{F}_3)$ consists of 24 elements.

### A.4 Experimental methodology

The $q$ and $l$ values used by the Ramanujan Graph construction for the convolutional layers as mentioned in Section 4.5 for various architectures is provided in Table 7.

Table 7: Values of $q$ and $l$ to generate Ramanujan graphs for layers of VGG, AlexNet and ResNet

| VGG13 | | | VGG19 | | |
|---|---|---|---|---|---|
| Conv Size | $q$ | $l$ | Conv Size | $q$ | $l$ |
| $[256 \times 256 \times 3 \times 3] \times 1$ | 13 | 177 | $[256 \times 256 \times 3 \times 3] \times 3$ | 13 | 177 |
| $[512 \times 256 \times 3 \times 3] \times 1$ | 19 | 121 | $[512 \times 256 \times 3 \times 3] \times 1$ | 19 | 121 |
| $[512 \times 512 \times 3 \times 3] \times 3$ | 19 | 242 | $[512 \times 512 \times 3 \times 3] \times 7$ | 19 | 242 |
| Conv to Linear Size | $q$ | $l$ | Conv to Linear Size | $q$ | $l$ |
| $2448 \times 25088$ | 47 | 533 | $2448 \times 25088$ | 47 | 533 |
| AlexNet | | | ResNet18 | | |
| Conv Size | $q$ | $l$ | Conv Size | $q$ | $l$ |
| $[384 \times 256 \times 3 \times 3] \times 1$ | 19 | 121 | $[64 \times 64 \times 3 \times 3] \times 4$ | 7 | 82 |
| $[384 \times 384 \times 3 \times 3] \times 1$ | 19 | 181 | $[128 \times 64 \times 3 \times 3] \times 1$ | 11 | 52 |
| $[256 \times 384 \times 3 \times 3] \times 1$ | 13 | 265 | $[128 \times 128 \times 3 \times 3] \times 3$ | 11 | 104 |
| | | | $[256 \times 128 \times 3 \times 3] \times 1$ | 13 | 88 |
| | | | $[256 \times 256 \times 3 \times 3] \times 3$ | 13 | 177 |
| | | | $[512 \times 256 \times 3 \times 3] \times 1$ | 19 | 121 |
| | | | $[512 \times 512 \times 3 \times 3] \times 3$ | 19 | 242 |
| Conv to Linear Size | $q$ | $l$ | | | |
| $2448 \times 25088$ | 47 | 533 | | | |

