# OpenReview forum: "Sparse Neural Architectures via Deterministic Ramanujan Graphs"
_TMLR — Accepted by TMLR_

### Review · Reviewer_j1B5 · 2024-07-21

**Summary Of Contributions:**

As far as I can see, this paper proposes a way to construct sparse neural architectures via the theory of Ramanujan graphs. Each neural layer is represented as a bipartite graph where the partition of vertices consists of the set of nodes of the layer and the set of nodes of the previous layer. Bi-regularity is assumed to ensure connectivity properties. These kinds of layers as described are stacked to produce a deep neural architecture which is sparse by construction. It appears that the proposed construction ensures path connectivity and trainability, which is one of the main claims. Another important claim is that these sparse networks upon training achieve comparable accuracy on classification tasks compared to the corresponding 'fully dense' networks. The numerical experiments demonstrate these claims for several neural architectures on CIFAR-10 and CIFAR-100 datasets, with comparisons to previous methods for constructing sparse networks and pruning. Importantly, the proposed approach is complementary to the literature on pruning at initialisation, since sparsity is achieved *before* training; as opposed methods that achieve sparsity by pruning a trained neural network.

**Audience:**

Yes

**Claims And Evidence:**

No

**Requested Changes:**

**Comments on the whole**
- The paper needs to be re-written, with a machine learning readership in mind.
- The abstract needs to be re-done to make it (1) useful, in the sense of highlighting the most important aspects of the paper, (2) aligned with the paper, in the sense of representing faithfully what's done in the paper, and (3) accurate and sound, as in being free of errors or misconceptions which may cause confusions.
- As mentioned already, the introduction section needs a major makeover to clearly frame the problem setting, motivation for the problem, approach and main results, connection with other works and so on. Make it helpful for readers to understand what the paper is about, why it is a relevant, what unsolved problems or obstacles it overcomes.
- Similar comments for section 2 which tries to set the lay of the land (research gap) and the contributions. The current list of contributions does not clearly describe what this paper achieves. Keep the list of contributions focused on what this work achieves. Put comparisons to other previous works separately, outside the list of contributions.
- I think casting this paper within the context of pruning at initialisation is inaccurate and potentially could lead to confusions. The paper is about a method to produce (create, define) sparse neural networks via the theory of Ramanujan graphs. As far as I understand, sparsity of the neural net is achieved by the proposed construction, not by pruning the counterpart 'fully dense' neural net (by counterpart meaning both neural nets have the same sets of nodes). This needs to be clarified. It would be okay to say that this work is complementary to the literature on pruning at initialisation.
- In section 3, interesting read about Ramanujan graphs, and discussions about guaranteeing rigorous Ramanujan, or that the networks remain Ramanujan; Cayley and so on. These things might be interesting *per se* for a mathematics readership, but for a machine learning readership it is important to explicitly establish the connections with (and implications for) machine learning, theory and practice.
- A comment on the intended meaning of *initialisation*: I get the impression that the authors conflate 'initialisation' with 'masking' and this confusion needs to be cleared. The way I understand these things, say there is a 'fully dense' neural architecture, then masking defines a subnetwork by setting some weights to zero. This I think is what you have called "network topology" in some other parts of the paper. While initialisation is the way to set the initial value of connection weights for a given architecture (a given topology). Then in principle these things are not the same, and this needs clarification in the paper to avoid confusions. Perhaps the authors had in mind some initialisation schemes which favour sparsity? Or initialisations designed for sparse networks?
- Section 4: This section heavily gives the impression of being written for mathematicians. I don't object mathematical definitions and abstractions when they serve a clear purpose for supporting the presentation of the method(s). Try keeping in mind a machine learning readership; then with this in mind, the mathematical definitions/abstractions need to be motivated and clearly connected with ML. Try writing notes to help readers: Tell readers that next they are going to see the definition of such and such, and why the definition is relevant for the machine learning problem that the paper tackles. Then, after the definition is presented, add short discussions (even at the expense of repetition) connecting the mathematical object just defined with corresponding parts of the neural network architecture, or whatever purpose the mathematical thing serves in the paper. The whole point being to avoid the style that appears to present mathematics for the sake of mathematicians, but rather to present for a machine learning readership.
- Regarding section 5, my main comment is this: I think the paper is missing sufficient details that are necessary for others to reproduce the results. The authors need to fill in the missing details about the step by step description of their method (Section 4) and how it is implemented in the experiments. Think of all the details that anyone would need in order to implement the method and obtain the numbers shown in the tables. Is the code going to be made available? Wy not make it available during the review?
- In connection with section 5, I reiterate once again my impression that there is a confusion between the meanings of initialisation, masking and pruning. I ask the authors to clarify whether they are in fact doing pruning (e.g. masking applied to a 'fully dense' architecture), or whether as I believe their method produces a sparse network by construction without needing pruning.
- References: The titles need to be capital-protected to read e.g. Cheeger, Cayley, Ramanujan (multiple occurrences), Alon-Boppana and so on. The roman numerals in the papers of Marcus, Spielman & Srivastava should be capitals. Paper of Noga Alon & Vitali Milman the title should read $\lambda_1$ not $\lambda$1. Check carefully the whole list of references to correct these flaws in paper titles. Also check publication venue names and use consistent format as much as possible.

**Page by page feedback**

Here I write comments for each page of the paper.

**Page 1**
- An inaccuracy to rectify in the abstract: I don't think you "present a sparsely connected, neural network architecture" but rather you present a method to construct such kind of architecture. The method is based on the theory of Ramanujan graphs.
- Also in the abstract: the reference to winning lottery tickets appeared all of a sudden without context. Perhaps mention the lottery ticket hypothesis before mentioning winning lottery tickets? And better if describing what a wining lottery ticket is in this context. If this is cumbersome, then perhaps avoid mentioning wining lottery tickets in the abstract.
- First sentence of the introduction is a bit vague. The parameter parsimony (funky way to refer to sparsity) is advantageous in various ways. Reducing training time is one and provided that the sparse network is trainable. Another one is computation cost of a forward pass of the trained network, as in when you want to use it for prediction on a new instance.
- Perhaps the most accurate way to frame your work is that it enables construction of a sparse network that is guaranteed to be trainable and so on. In general, claims presented here need to be aligned with what is done in the paper.
- I would suggest deleting "backbone" and instead write e.g. 'fully dense' network.
- I suggest avoiding the acronym "PaI" and instead write "pruning at initialisation" in words.
- Explain the meaning of "multi-shot" in this context, as per the intended meaning.
- Then explain the meaning of "zero-shot" in this context.
- Replace "an initialization topology" with something that better describes what's intended (perhaps "an initial network topology"?)
- Last sentence of first paragraph: merge it into the second paragraph.
- First sentence of third paragraph: replace "a sparse initialization architecture" with "a sparse initial architecture"? Or change as needed to clarify the intended meaning.
- Previous to last sentence of this paragraph: the meaning is almost completely obscure to me.
- Last sentence: write something more straightforward than "arrest" according to the intended meaning.

**Page 2**

- Top: The meaning of "initialization" is again not what is usually called initialization in the literature on neural networks. Change this to describe more accurately what's intended. Or define what you mean by "network initialization" (which could conflict with the usual meaning of this, but at least a definition would make it clear what you are trying to refer to).
- Why is it important to adhere to the rigorous definition of Ramanujan graphs? What's the relevance for ML?
- Maybe write 'fully dense' network instead of "dense network"
- Description of the organisation of the paper: insert section numbers to improve.
- Section 1.1 on related works: Appears minimal. Perhaps that's okay but at least acknowledge that you mention only a minimal sample of the extensive literature. Some important missed related works include [1], [2] (see below). If any technical term is mentioned here, its meaning needs to be explained. For instance, what is "one-shot neural network pruning"
- Fist paragraph of section 2: Not clear how the zero-shot implies advantage in generalization capabilities, this needs at least some reference or further comment explaining.
- Regarding (i): Again, why is ensuring Ramanujan property relevant? Try to improve the connection with ML.
- In (ii): As suggested, avoid "PaI" and write explicitly "pruning at initialization" instead.
- In (iii): Is "symmetricity" a word? I think not. Replace with "symmetry" which is.
- Claim that "This is the first [..]" needs to be accompanied by a comment explaining why the machine learning readership should pay attention to this.

**Page 3**

- Figure 1: In the caption, better write "An example of two consecutive neural layers where each layer is a bipartite Ramanujan graph." To improve the figure, at the top of the columns of nodes put the labels $I$, $J$ and $K$.
- Contribution 1: In the absence of a definition of the meaning of "zero-shot pruning" earlier in the paper, this contribution is difficult to understand. Also, move "Previous [..]" to somewhere outside the list of contributions, perhaps after it in a new paragraph discussing these connections and comparisons to previous works.
- Contribution 2: Change "initializing" with something that better describes the intended meaning.
- Contribution 3: Move "Previous [..]" to somewhere outside the list of contributions, perhaps after it in a new paragraph discussing these connections and comparisons to previous works.
- Contribution 4: The way it is written, it does not appear to be describing a contribution at all. Perhaps this is something that also needs to be moved to somewhere outside the list of contributions.
- Contribution 5: This appears to be clarifying that your method works for both kinds of layers, fully connected and convolutional, so this is not a separate contribution but rather a clarification point to add to contribution 1 or contribution 2?
- Two-line paragraph after the list of contributions: Unclear what's being claimed here. Try expanding and clarifying as per the intended purpose of this sentence. Good place to elaborate on the contributions, including the content that I suggested to be moved outside the list of contributions.
- First section of section 3: This is again mentioning the not guaranteeing being rigorously Ramanujan, but missing the connection of this point with neural network properties that the machine learning readership would/should care about.
- Again, clarify the meaning of "network initialization" and "initialized networks" or reformulate to better describe and avoid the conflation with what's commonly called "initialization" in the literature on neural networks.
- Paragraph at the bottom: For readers to understand this, somewhere earlier in the paper it should have been mentioned that each neural network layer corresponds to a bipartite graph, that the whole deep network consists of stacking (concatenating?) such bipartite graphs and so on. Then this paragraph would be focusing on clarifying the point that regularity assumption ensures path-connectedness.

**Page 4**

- Replace "High-symmetricity" with "High symmetry"
- At least mention a lay terms description of $\mathbf{Z}_5$, that it is the finite group with modulo 5 operations.
- Related note: Currently it is unclear what's the role of the mathematical technicalities mentioned here about this finite group, generating set, adjency matrix and so on. Could the authors elaborate on the connection to neural networks.
- Section 4: As I commented before, this section currently appears to be written for a mathematics readership, and the request is to improve it adding comments, pointers, motivations, clarifications etc. to improve the connection to neural networks for the sake of the machine learning readership.
- A special case of this is the paragraph before section 4.1: What should machine learning readers get from it? Why should they care?

**Page 5**

- Figure 2: In the caption, better write "An example of two consecutive neural layers where each layer is a biregular bipartite Ramanujan graph." To improve the figure, at the top of the columns of nodes put the labels $I$, $J$ and $K$.
- Perhaps change the notations for vertex Cheeger constant and edge Cheeger constant, using suggestive notation that is easy to hold in the mind? Would the authors consider something like $\mathbf{h}_v(\Gamma)$ and $\mathbf{h}_e(\Gamma)$?
- A more important point to resolve is why are these Cheeger constants appearing here? Do they play a crucial role for what's coming next, especially for the description of the method that the machine learning readers would need to have these constants in mind while reading about the method and demonstrations in the experiments and so on?
- The connections between these mathematical abstractions and the machine learning problem targeted in this paper (I think it is constructing sparse neural architectures) need to be made explicit and clear.
- Bottom of the page: Insert a clickable cross reference to the corresponding appendix for the Cheeger-Buser inequality.

**Page 6**

- Top of the page: Insert a clickable cross reference to the corresponding appendix for the  Alon-Bopanna theorem.
- Improve the motivation for the need for Definition 4.4 (quadratic residue, Legendre symbol). Even better if the authors add after the definition a short paragraph elaborating on this and its connection with what is to come next.
- Similar comments for Definition 4.5. And use $\mathbb{K}$ here to denote a generic field.
- Displays defining $GL_2(\mathbb{K})$ and $Z(\mathbb{K})$: delete "(in $\mathbb{K}$)" and add a line after the second display saying something to the effect that "In these definitions it is assumed that all entries are elements of the field $\mathbb{K}$."
- Remark after Definition 4.5: What is $\mathbb{F}_q$? Is it not $\mathbb{Z}_q$?
- First line of section 4.2: Ensure "$\mathrm{mod}$" instead of the current "$\mathit{mod}$ " (twice)
- It is necessary to explain with copious details, with the machine learning readership in mind, the meaning of everything in points 1, 2, 3, 4 describing the construction. Is 1 a step of the construction and not just a definition? In step 2, a lot of things need to be explained, especially the intriguing $i^2 = -1 \pmod q$ when you are supposedly working with finite fields. Step 3 requires definitions of the meaning of generating set and the notation "Cay" that appears here for the first time in the paper. Step 4 appears to be stating a theorem or proposition, so I am not sure how to see it as a step in a method to construct sparse networks.

**Page 7**

- Section 4.3 does not contain nearly enough information for others to understand how the construction of fully connected layers goes.
- Section 4.4 needs to improve with comments on connections/implications for the sparse neural networks.
- Section 4.5 does not contain nearly enough information for others to understand how the construction of convolutional layers goes.

**Page 8**

- Section 4.7 appears to be of interest for graph theory or mathematics readership. The connection with ML and NNs is missing.
- First line of section 5: The implied meaning of "effectiveness" is unclear. What metric is being implied here?
- Similar comment for "robustness" a few lines down.
- Once again, try to clarify whether the method is pruning or not. Perhaps the claim is that pruning is implicit by the creation of the sparse neural Ramanujan graph layers by means of the proposed method (but the steps of the proposed method are unclear).
- "The performance of the 'fully dense' and the pruned networks"
- Once again, write "pruning at initialization" instead of "PaI"
- Is the code going to be made available?

**Page 9**

- "corresponding 'fully dense' networks."
- Section 5.3: For understanding the descriptions, it is necessary once again that Section 4 gives an exposition, with copious details, of the proposed constructions for fully connected layers and for convolutional layers.
- Paragraph after Table 3: "the effective number of connections being equal to"

**Page 10**

- Replace "upto" with "up to"
- Table 4: Replace "(Remaining Edge Percentage)" with "(Network Density)" for both tables.
- In the paragraph before this table: "summarized in Table 4. By definition, network density is the remaining edge percentage. It can be observed [..]"
-  "construction allows us to extremely prune the fully connected layer to a network density of 0.43% while still retaining [..]"
- Paragraph after Table 4: "achieve an overall network density of" (replace "pruning percentage" with "network density")
- Paragraph after Table 5: Write "pruning at initialization" instead of "PaI" (twice)

**Page 11**

- Section 6 (conclusion) is rather vague currently. There is space to expand and improve.

**Global changes**
- Replace "symmetricity" with "symmetry"
- Replace "PaI" with "pruning at initialisation" (make explicit, everywhere).
 - Maybe write 'fully dense' network instead of "dense network"
- Replace "convolution layer" with "convolutional layer" as per the usage in the neural networks literature.

**Missed literature**

[1] Soufiane Hayou, Jean-Francois Ton, Arnaud Doucet & Yee Whye Teh. Robust Pruning at Initialization. ICLR 2021.

[2] Laurent Orseau, Marcus Hutter, Omar Rivasplata. Logarithmic Pruning is All You Need. NeurIPS 2020.

**Strengths And Weaknesses:**

**Strengths**
- The problem (sparse and trainable/performant neural architectures) is relevant for TMLR.
- Appears to contribute results that merit attention of the readership, both 'theory' and empirical evaluations.
- Interesting to see the constructed sparse networks exhibiting high sparsity (small density) while retaining trainability and good post-training performance in terms of accuracy compared to their 'fully dense' counterparts.

**Weaknesses**
- Overall the presentation and organisation of the paper need improvement.
- The introduction section needs a major makeover to clearly frame the problem setting, approach and so on.
- There are some misconceptions, e.g. conflating initialisation with network topology (connectivity structure), and other.
- The paper appears to be written for mathematicians, esp. sections 2, 3, 4 which are heavy in mathematical abstractions, with little or no exposition of their connection with neural network architectures.
- Crucially, the step by step description of the method to construct the sparse networks is not nearly clear enough.

---

> ### Author Response · Authors · 2024-09-12
> **Revised version: Rewritten the abstract, introduction, added motivation for mathematical concepts for the ML audience, more details on the construction methods are also added.**
>
> We thank the anonymous reviewer for the detailed and constructive review which has helped us a lot to improve the manuscript. Below we detail the changes made.
>
> ### Reply to General Comments:
>
> 1. Almost all the sections have been rewritten by including more explanation of the mathematical notions. The machine learning perspective is also mentioned in the relevant sections.
> 2. The abstract has been rewritten keeping in mind the clear and valuable suggestions from the reviewers.
> 3. We have completely rewritten the Introduction section highlighting the problem setting, motivation, approach, and results. We sincerely thank the reviewers for the excellent suggestions.
> 4. We have rewritten the research gap and contributions by directly focusing on the methodology proposed in the paper. We thank the reviewers for the valuable suggestions.
> 5. These comments has been very useful for us in recasting the paper. We believe that now have a much better and direct presentation of the proposed approach and its contribution.
> 6. We have added the discussion on the connection between Ramanujan graphs and neural networks.
> 7. Masking is the intended operation to sparsify the network. We appreciate the clarity provided by the reviewer.
> 8. The sections has been thoroughly rewritten.
> 9. The code will be made available after the review cycle ends since this is the first ever construction of Ramanujan graphs to the best of our knowledge.
> 10. We have rewritten the section to remove the confusion. Our method produces a sparse network by construction. It nnever does pruning or masking.
> 11. We have corrected the errors in References.
>
> ### Reply to Page by Page Feedback:
>
> #### Page 1
>
> 1. We have rewritten the abstract to reflect the proposed methodology more faithfully.
> 2. We no longer mention lottery tickets in the abstract.
> 3. We have rewritten the Introduction.
> 4. We thank the reviewers for the valuable suggestion. We have rewritten the Introduction to highlight this. We do get a lot more clarity on the contributions of the paper due to this comment.
> 5. We have made this change.
> 6. We have made this change.
> 7. In this context, multi-shot means iterative pruning. We have removed the usage of this word in our text.
> 8. In our context, zero-shot means non-iterative generation. We borrowed this term from the pruning literature. We have now removed the usage of this word.
> 9. We have edited the text to reflect this change.
> 10. We have significantly rewritten this section.
> 11. We have replaced the phrase.
> 12. We have rewritten the paragraph.
> 13. Thank you for the suggestion. We have rewritten the sentence.
>
> #### Page 2
>
> 1. We have edited the text to clarify this. We intend to mention here the sparse topology which will be initialized with random weight values and then trained on the data.
> 2. We have discussed this in the text.
> 3. We have made this change.
> 4. We have inserted the section numbers.
> 5. We have majorly rewritten the related works section. We only provide a snapshot of the literature since the entire literature is too voluminous on this topic. The mentioned references are now added.
> 6. We have rewritten the section. This sentence is no longer stated.
> 7. The Ramanujan property ensures path connectivity, which is important for the forward pass as well as backpropagation in a neural network. This is now mentioned in Section 4.
> 8. Thank you for the suggestion. The change has been made at all occurrences.
> 9. We have replaced "symmetricity" with "high symmetry."
> 10. The importance of these properties in terms of neural networks is now explained in Section 4.
>
> #### Page 3
>
> 1. The caption is updated.
> 2. The contributions have been rewritten to reflect this point.
> 3. We have rewritten the contributions.
> 4. We have rewritten the contributions.
> 5. We have rewritten the contributions.
> 6. We have rewritten the contributions.
> 7. We have rewritten the contributions.
> 8. We have rewritten this section to highlight the connection between these graph-theoretic properties and neural network learning.
> 9. We have rewritten this section to clarify this confusion.
> 10. This point is now mentioned in the Introduction section.
>
> #### Page 4
>
> 1. We have made this change.
> 2. This has now been detailed.
> 3. This has been clarified at the end of Section 3.
> 4. Further details have been added to make it more readable for a neural network readership.
> 5.  Further details have been added to make it more readable for a neural network readership.

---

> > ### Author Response · Authors · 2024-09-12
> > **Response to Page by Page Feedback (continued)**
> >
> > #### Page 5
> >
> > 1. The figure and the captions have been updated.
> > 2. The two (vertex and edge) Cheeger constants are now denoted by \( \mathbf{h}_V \) and \( \mathbf{h}_E \) respectively.
> > 3. The motivations are now added for establishing the notions of Cheeger constants in Section 4.1.1.
> > 4. This has now been made explicit in Section 4.1.1.
> > 5. This has now been corrected.
> >
> > #### Page 6
> >
> > 1. This has now been changed.
> > 2. The motivation has been added.
> > 3. This has now been changed.
> > 4. This has now been changed.
> > 5. By $\mathbb{F}_q$ we mean the finite field of $q$ elements ($q$ is a prime power). In general, it is not $\mathbb{Z}_q$ (the integers modulo $q$), because $\mathbb{Z}_q$ doesn't form a field if $q$ is not a prime, e.g., $\mathbb F_9$ is not $\mathbb Z_9$. However, in this work, we are taking $q$ to be a prime, so one can interchange.
> > 6. This has now been changed.
> > 7. This portion has been updated with more details and motivation.
> >
> > #### Page 7
> >
> > 1. Section 4.3 is rewritten with more details.
> > 2. The section has been rewritten.
> > 3. The section has been rewritten.
> >
> > #### Page 8
> >
> > 1. The section has been rewritten.
> > 2. Effectiveness intended to mean accuracy. The section has been rewritten.
> > 3. The section has been rewritten.
> > 4. We thank the reviewers for pointing out this confusion. We have rewritten the section significantly to highlight the intended meaning.
> > 5. The sentence has been revised.
> > 6. The change has been made.
> > 7. The code will be made available once the review cycle ends and will be published on our GitHub pages.
> >
> > #### Page 9
> >
> > 1. We have changed the sentence to clarify the meaning of "corresponding fully dense" networks.
> > 2. We have expanded this section by including more details about the methodology.
> > 3. The meaning of "effective number of connections" is clarified in this paragraph in the revised text.
> >
> > #### Page 10
> >
> > 1. The change has been made.
> > 2. The change has been made.
> > 3. We have rewritten the sentence.
> > 4. We have made the change.
> > 5. We have made the change.
> > 6. We have made the change.
> >
> > #### Page 11
> >
> > 1. We have rewritten the conclusion to highlight the contribution of the paper more clearly.
> >
> > #### Global Changes
> >
> > 1. We have made the change.
> > 2. We have made the change.
> > 3. We have made the change.
> > 4. We have made the change.
> >
> > #### Missed Literature
> >
> > Thank you for suggesting the important references that we missed. We have included them in the revised paper.

---

> ### Comment · Reviewer_j1B5 · 2024-09-24
> **Response to the rebuttal and the updated version**
>
> Thanks to the authors fort he revised version and for welcoming my feedback.
>
> I only have minor comments:
>
> - Title: Please consider replacing "and" and "via"
> - Section 1.1, first sentence: "Sparse neural networks have been" (notice "have")
> - A few lines down: "$L_0$ regularization based training which encourages zero weights has been" (notice "has")
> - "Initial research works in this direction were based on applying" (add "works")
> - "Robust methods of pruning at initialization have been proposed" (notice "have")
> - Section 2, first paragraph: "This does not always guarantee path-connectedness" (notice "does")
> - Page 3, paragraph on High-symmetry: "The adjacency matrices of Cayley graphs and of covers of Cayley graphs have much more
> symmetry than those of general regular graphs."
> - A bit further down: Try displaying the matrix $Adj$ and matrix $B$
> - "The adjacency matrix of a graph captures the structure" (note "captures")
> - Page 4, top: "need to be" (notice "need")
> - "ensures that the initialized networks remain Ramanujan, are path-connected and are highly symmetric, being either Cayley graphs of certain algebraic groups to replace the balanced dense bipartite graphs, or the Ramanujan r-covering of full bi-regular bipartite graphs to replace the unbalanced dense bipartite graphs." (notice the commas)
> - Section 4, in item 2: "These graphs are used to define the neural network layers." (avoid "modelise")
> - Paragraph after Definition 4.1: Replace "transforms" with "reduces"
> - Next paragraph: "We shall define our pruned network according to these constructions." (avoid "modelise")
> - Next page: "This allows for free flow of information throughout the network which the graph defines." (avoid "modelises")
> - Section 4.1.2: "Given a finite undirected graph $\Gamma$, the eigenvalues"
> - "(A graph is said to be d-regular if there are exactly d-edges attached to a vertex.)"
> - Note that "bounded away from zero" means $\geq c > 0$.
> - Is it not enough to say that both these quantities are positive ($>0$) in this passage?
> - Next page, Definition 4.4: "integer $q$ is called a quadratic residue modulo $n$ if it is congruent to a perfect square modulo $n$, i.e., if there exists an integer $x$ such that $x^2 ≡ q \pmod n$."
> - Also: The final "n" should be "$n$"
> - Remark on page 6, just before Section 4.2: $PGL_2(\mathbb{F}_q)$ (twice).
> - Section 4.2, first line: "(the condition of $\equiv 1\pmod 4$ can be" ?
> - Fix " (Lubotzky et al. (1988)." (close parenthesis missing)
> - Page 7: The notation "$Cay(PGL_2(\mathbb{F}_q),S)$" needs to be specified. (Also, make sure the notation used in the main paper matches the notation used in the Appendix.)
> - Page 9, Section 4.7: "Further, they showed " (notice the comma)
> - Section 5, first line: Replace "initialization" with "construction" ?
> - CIFAR-10, CIFAR-100
> - Page 10, Section 5.2, top: "The performance of the pruned networks is compared"
> - Further down, in Section 5.3: "For the second part, we consider" (notice the comma)
> - "is given in Table 3, while for the rest of the architectures" (notice the comma)
> - Page 11: CIFAR-10, CIFAR-100 (several times here, and do this in other pages as well)
> - " The accuracy of the Ramanujan graph based sparse networks remains close" (notice "remains")
>
> Appendix:
> - A.2.5: "The existence and construction of expanders are deep questions and those of Ramanujan graphs are even
> deeper."
> - Make sure that notations match between the main paper and the appendix. For instance, $PGL_n$ (in the main part) vs $\mathrm{PGL}_n$ (in the appendix), and similar for GL and Z.
>
> Throughout the paper:
> - Choose one of "bipartite" or "bi-partite" and use consistently. (Preference for "bipartite")
> - Choose one of "biregular" or "bi-regular" and use consistently. (Preference for "biregular")
> - Insert in the main part cross-references to the parts of the appendix where things are defined, more details and so on.
> - Check the use of textual "Author (Year)" format (with \citet) and parenthetical "(Author, Year)" format (with \citep) for citations, and straighten it out using the appropriate format in each passage of the paper.

---

> > ### Author Response · Authors · 2024-11-03
> > **Response to further corrections**
> >
> > Thank you for your detailed review and constructive feedback. In this second revised version we addressed each of the points mentioned and incorporated the suggested changes. We apologise for the delay as we were not sure about the timeline.
> >
> > 1. Title: We have replaced "and" with "via" in the title, as suggested.
> >
> > 2. Section 1.1, first sentence: This has been incorporated.
> >
> > 3. Section 1, a few lines down: This has been incorporated.
> >
> > 4. Initial research works: The phrase has been amended to "Initial research works in this direction were based on applying."
> >
> > 5. Robust methods of pruning: This has been incorporated.
> >
> > 6. Section 2, first paragraph: This has been incorporated.
> >
> > 7. Page 3, paragraph on High-symmetry: This has been incorporated.
> >
> > 8. Section 2, matrix notation: We have clarified the distinction by displaying the matrix Adj and matrix B separately.
> >
> > 9. The adjacency matrix of a graph: This has been incorporated.
> >
> > 10. Page 4, top: This has been incorporated.
> >
> > 11. Ensures that the initialized networks remain Ramanujan: We restructured this sentence, incorporating the commas as advised.
> >
> > 12. Section 4, item 2: modelise has been changed to define
> >
> > 13. Paragraph after Definition 4.1: This has been incorporated.
> >
> > 14. Next paragraph: modelise has been changed to define
> >
> > 15. Following page: modelise has been changed to define
> >
> > 16. Section 4.1.2: This has been incorporated.
> >
> > 17. A graph is said to be d-regular: This has been incorporated.
> >
> > 18. We have changed it to positive.
> >
> > 19. Yes, we have changed it now.
> >
> > 20. Next page, Definition 4.4: This has been incorporated.
> >
> > 21. "n" in final notation: This has been incorporated.
> >
> > 22. Remark on page 6, just before Section 4.2: This has been incorporated.
> >
> > 23. Section 4.2, first line: This has been incorporated.
> >
> > 24. Fix "Lubotzky et al. (1988)": This has been incorporated.
> >
> > 25. Page 7 notation \( Cay(PGL_2(\mathbb{F}_q), S) \): This has been incorporated.
> >
> > 26. Page 9, Section 4.7: This has been incorporated.
> >
> > 27. Section 5, first line: Initialization replaced with construction.
> >
> > 28. CIFAR-10, CIFAR-100: Ensured uniform formatting of these dataset names.
> >
> > 29. Page 10, Section 5.2, top: This has been incorporated.
> >
> > 30. Further down, in Section 5.3: This has been incorporated.
> >
> > 31. Table 3 citation: This has been incorporated.
> >
> > 32. Page 11, CIFAR-10, CIFAR-100: Ensured uniform formatting of these dataset names.
> >
> > 33. The accuracy of the Ramanujan graph-based sparse networks: This has been incorporated.
> >
> > Appendix:
> >
> > 1. Appendix, A.2.5: This has been incorporated.
> >
> > 2. Consistency of notation in the appendix: The notations have been harmonized between the main text and the appendix.
> >
> > Throughout the paper:
> >
> > 1. We are using bipartite throughout now.
> >
> > 2. We are using biregular throughout.
> >
> > 3. Cross referencing and more details especially those with respect to Appendix A.3 Projective Linear Groups have been introduced.
> > 4. We are now using \citep
> >
> > Thank you again for your valuable input.

---

### Review · Reviewer_FLZe · 2024-08-06

**Summary Of Contributions:**

The authors present a sparse neural network approach using Ramanujan graphs, taking into account properties like path connectivity and symmetricity. The authors show that the method provide comparable accuracy and sparsity with competing pruning approaches.

**Audience:**

Yes

**Broader Impact Concerns:**

No ethical concerns.

**Claims And Evidence:**

Yes

**Requested Changes:**

The general theme of using spectral graph theory and related CS theory techniques for sparsification of neural networks is interesting. Relevant work by Tam and Dunson (https://arxiv.org/pdf/2304.03096) look at sparsification of neural networks from a spectral gap/regularization perspective that I think would be interesting for the authors to comment on.

**Strengths And Weaknesses:**

Strengths:

Clarity: The paper is written in a clear and easy to follow manner.
Quality: The quality and scope of the experiments are sufficiently broad, and they generally support the authors claims. The quality of any mathematical derivation is clear and to the best of my knowledge, sound.
Novelty: The proposed approach is, to the best of my knowledge, useful.

Relevance to TMLR and audience: I believe that the content and topic of the paper fits the scope of TMLR, and fits the criteria that at least a portion of the audience of TMLR will find this article interesting.

---

> ### Author Response · Authors · 2024-09-12
> **Revised version: incorporated the useful reference**
>
> We thank the anonymous reviewer for the constructive review and for pointing out this interesting article. We have incorporated it in sec 1.1 Related Work. Tam and Dunson use the second smallest eigenvalue of the weighted Laplacian matrix (modified by taking the magnitude of the weights) in the objective function of the neural network training as a tool for regularization taking into account the connectivity. It will be interesting to see what happens if we run this technique of regularisation on top of our proposed architecture.

---

### Review · Reviewer_g8Qr · 2024-09-05

**Summary Of Contributions:**

The authors define an algorithm for constructing a bipartite graph that is Ramanujan. These have theoretical properties that encourage high 'connectivity' (in an expander sense) while being very sparse. This algorithm is applied to construct weight masks for fully connected and convolutional layers, creating sparse neural networks. Unlike other techniques, this method is deterministic and never looks at the data.

**Audience:**

Yes

**Claims And Evidence:**

No

**Requested Changes:**

Critical
- I believe a thorough rewrite with a Machine Learning audience in mind, in particular of the abstract and first couple of sections, is necessary.
- Hoang 2023 seems like a paper that's very close in both goal and approach, as it also aims to construct Ramanujan graphs. The work currently doesn't make clear what the difference is. Also I wasn't able to follow the argument about Hoory 2006 related to this.
- The Appendix A.1 and A.2 contains some useful definitions that help with understanding the paper, however the main paper rarely links to them. This would significantly help the reader with understanding.
- Although the paper is fairly self-contained with background in the appendix, it is missing a lot of background on (projective) linear groups and finite fields. I was not able to understand the construction of PGL_2(F_q). A small example would significantly help
- The constructions in section 4.2 and 4.4 lack a clear example. It would significantly help the understanding if the steps are explained with a running example.
- Where does the construction in 4.2 come from? Lubotzky 1988? This is currently unclear.
- The time complexities given in 4.6 do not have a proof or reference.
- Why are there no confidence bounds / standard deviations on the experiments? The numbers are close together, so it could be due to randomness.

Improvements
- Several places uses the term 'multi-shot'. I assume this means data dependency?
- Symmetry, as discussed in the context of Cayley graphs, isn't well defined. It is shortly mentioned that symmetry helps for computational purposes, but this could be elaborated on a lot more.
- Section 3: I tried to reproduce the Cayley graph with the generating set S={(1, 0), (1,1), (1,4)}, but I got a different graph than shown in Figure 1. It's possible I misunderstood, but it seems like the generating set is (1, 0), (1,1), (1, 2) or something? But this isn't invertible.
- 4.1.2: "The quantities are bounded away from zero". It wasn't clear to me whether this is > or =>
- Appendix A.2.3: The quantity d doesn't seem to be defined anywhere
- Definition 4.5: Mention PGL is the projective linear group
- Remark below: \mathbb F_q is never defined.
- There is no intuition, proof or description why the constructions in 4.2 and 4.4 lead to Ramanujan graphs, except for a reference. This could significantly help understanding.
- 4.2: It is not clear to me why a complex valued matrix would be an element of this linear group on finite fields.
- 4.3: "The Ramanujan graph structure of LPS". What is LPS?
- Why is SynFlow not applied to the Resnet?

**Strengths And Weaknesses:**

Strengths: I learned a lot: there are some concepts from graph and group theory that I was unfamiliar with that are quite decently explained in this paper (although I did regularly need Wikipedia or ChatGPT for more intuition). The method has interesting graph-theoretic guarantees, and seems to perform quite well compared to existing methods experimentally, although I'm not knowledgeable enough in this field to judge properly.

Weaknesses:
This paper reads more like a mathematics paper than a paper on sparse neural architectures. This need not be a problem - but I suspect a large part of the intended audience may shy away from an abstract and introduction that's this heavy on the presumption that the reader is aware what Ramanujan, Cayley, and bi-regular bipartite graphs are, for example. I think the body of the text, with some adjustments, is reasonably understandable, but the technicality of the abstract and introduction can be toned down to significantly improve readability.

In particular, the introduction could do a better job by focusing rather on why the properties the algorithm induces are relevant to the task. Why is symmetricity important, for instance? Why is regularity important? What are expander networks, and what is the intuition that it improves empirical performance?

Furthermore, the main algorithm should be elaborated more with proper reference for its claims.
Finally, the experiments are on fairly small datasets and with small / old models. I'm again not familiar enough to judge if this is significant experimental evidence within the field.

---

> ### Author Response · Authors · 2024-09-12
> **Revised version: rewritten the paper incorporating the comments. Detailed changes are listed below**
>
> We thank the anonymous reviewer for the constructive review. Below are the responses and the changes made:
>
> #### Critical
>
> 1. The abstract, Introduction, Related Work, and Contribution sections are thoroughly rewritten, clearly stating the objectives and contribution of the paper.
> 2. Previous proposed approaches so far have led to the formation of irregular graph networks that do not strictly adhere to the rigorous notion of Ramanujan graphs. Hoory in [3] proposed criteria for irregular Ramanujan graphs, but additional criteria that none of the works check for exist. Hoang et al. (Hoang, Duc NM, et al. "Revisiting pruning at initialization through the lens of Ramanujan graph." ICLR 2023) do not give a pruning construction but rather perform an analysis of existing pruning methods (which result in irregular networks) via the lens of regular Ramanujan graphs by looking at the regular subgraphs within the irregular graphs. In our case, our initial graphs are regular and rigorously Ramanujan, avoiding the complications associated with irregularity.
> 3. The definitions provided in the Appendix are now linked in Section 4 of the main paper.
> 4. We have added background material on Projective Linear Groups in Appendix A.3. A running example is also added in Section 4.2.
> 5. A running example is added, and the explanation is enhanced for better understanding.
> 6. The construction in Section 4.2 comes from [1]. This has now been made clear.
> 7. The derivation of the time complexity is now elaborated.
> 8. We had repeated the experiment several times (around 4 to 5), and obtained results very similar to the ones we have reported in each run. Due to this, we did not provide any confidence bounds.
>
> #### Improvements
>
> 1. Indeed, this means data dependency.
> 2. We would like to take up these studies in our future work.
> 3. This is because in the figure we are using the following label mapping:
>    $i_r = (0,r-1), r=1,2,3,4,5, j_1 = (1,4), j_2 = (1,0), j_3 = (1,1), j_4 = (1,2), j_5 =(1,3).$
>    Your obtained graph will be isomorphic.
> 4. It means $>$. This has now been clarified.
> 5. The quantity $d$ has now been defined.
> 6. This has now been mentioned directly in the definition.
> 7. $\mathbb F_q$ has now been defined inside the definition.
> 8. Cayley graphs of finite simple groups are in general expanders (this follows from the Super strong approximation theorem). However, to show that the Lubotzky--Phillips--Sarnak [1] graphs are Ramanujan is quite involved. A nice exposition can be found in [2], Chapter 4. In the appendix, we give the reference from where the interested reader can get the proof. For the graphs presented in Section 4.4, we need to use the properties of cyclic shift permutation matrices to compute the eigenvalues directly.
> 9. Actually, $i$ is an integer solution to the equation $i^2 = -1$ in the finite field of $q$ elements. The matrix is not complex valued. It has now been clarified: where $i$ is some fixed "integer" solution to $i^2 = -1$ (mod $q$).
> 10. This has now been corrected. By LPS, we meant Lubotzky--Phillips--Sarnak [1].
> 11. SynFlow does not perform very well and has low accuracy with ResNet architectures, which is why we omitted it.
>
>
> [1] A Lubotzky, R Phillips, and P Sarnak. Ramanujan graphs. Combinatorica, 8:261–277, 1988.
>
> [2] Giuliana Davidoff, Peter Sarnak, and Alain Valette. Elementary Number Theory, Group Theory and
> Ramanujan Graphs. London Mathematical Society Student Texts. Cambridge University Press, 2003.
>
> [3] Shlomo Hoory. A lower bound on the spectral radius of the universal cover of a graph. Journal of
> Combinatorial Theory, Series B, 93(1):33–43, January 2005

---

### Decision · Action_Editor_mmCg · 2024-11-08

**Recommendation:** Accept as is

**Comment:**

The authors put in a considerable amount of work to improve the clarity and presentation of their work, which was one of the main criticisms of the original submission. All authors, and myself, are in agreement that after these improvements, the work is suitable for acceptance.

**Audience:**

After the latest revisions, the audience was broadened to a more general ML audience, as the authors have made the writing more accessible (thanks to suggestions from the reviewers). This work would be of particular interest to those working with sparse neural networks.

**Claims And Evidence:**

The claims made in the paper appear to be well-supported through both a theoretical presentation and empirical evaluation.